# Biceps Tenodesis Better Improves the Shoulder Function Compared with Tenotomy for Long Head of the Biceps Tendon Lesions: A Meta-Analysis of Randomised Controlled Trials

**DOI:** 10.3390/jcm12051754

**Published:** 2023-02-22

**Authors:** Chunsen Zhang, Guang Yang, Tao Li, Long Pang, Yinghao Li, Lei Yao, Ran Li, Xin Tang

**Affiliations:** 1Department of Orthopedic Surgery, West China Hospital, Sichuan University, Chengdu 610041, China; 2Orthopedic Research Institute, West China Hospital, Sichuan University, Chengdu 610041, China; 3Operating Room of Anesthesia Surgery Center, West China Hospital, Sichuan University, Chengdu 610041, China

**Keywords:** biceps tendon, tenotomy, tenodesis, meta-analysis

## Abstract

Purpose: Surgical options for long head of the biceps tendon (LHBT) lesions include tenotomy and tenodesis. This study aims to determine the optimal surgical strategy for LHBT lesions with updated evidence from randomised controlled trials (RCTs). Methods: Literature was retrieved from PubMed, Cochrane Library, Embase and Web of Science on 12 January 2022. Randomised controlled trials (RCTs) comparing the clinical outcomes of tenotomy and tenodesis were pooled in the meta-analyses. Results: Ten RCTs with 787 cases met the inclusion criteria, and were included in the meta-analysis. Constant scores (MD, −1.24; *p* = 0.001), improvement of Constant scores (MD, −1.54; *p* = 0.04), Simple Shoulder Test (SST) scores (MD, −0.73; *p* = 0.03) and improvement of SST (*p* < 0.05) were significantly better in patients with tenodesis. Tenotomy was associated with higher rates of Popeye deformity (OR, 3.34; *p* < 0.001) and cramping pain (OR, 3.36; *p* = 0.008]. No significant differences were noticed between tenotomy and tenodesis regarding pain (*p* = 0.59), American Shoulder and Elbow Surgeons (ASES) score (*p* = 0.42) and its improvement (*p* = 0.91), elbow flexion strength (*p* = 0.38), forearm supination strength (*p* = 0.68) and range of motion of shoulder external rotation (*p* = 0.62). Subgroup analyses showed higher Constant scores in all tenodesis types and significantly larger improvement of Constant scores regarding intracuff tenodesis (MD, −5.87; *p* = 0.001). Conclusions: According to the analyses of RCTs, tenodesis better improves shoulder function in terms of Constant scores and SST scores, and reduces the risk of Popeye deformity and cramping bicipital pain. Intracuff tenodesis might offer the best shoulder function as measured with Constant scores. However, tenotomy and tenodesis provide similar satisfactory results for pain relief, ASES score, biceps strength and shoulder range of motion.

## 1. Background

Long head of the biceps tendon (LHBT) lesions are highly prevalent pathologies causing anterior and deep shoulder pain, many of which are concomitant with rotator cuff tears [1,2,3,4,5]. Current nonoperative treatment for LHBT pathologies includes rest, icing, anti-inflammatory oral drugs or injections and physical therapy. Surgical interventions may be indicated if conservative treatments are not satisfactory. The most used surgical techniques are biceps tenotomy and biceps tenodesis. Tenotomy was believed to be easy and fast, with simple rehabilitation and may achieve similar pain relief and shoulder movement range when compared with tenodesis [6,7]. Moreover, it was argued that tenodesis minimises the risks of Popeye deformity and cramping bicipital pain, and better maintains muscle strength [7,8,9].

However, previous similar meta-analyses of randomised controlled trials (RCTs) [10,11,12,13] and RCTs plus cohort studies [14,15,16] reported controversial results regarding biceps and shoulder function. Moreover, none of the previous meta-analyses have compared the improvement in functional scores from baseline between tenotomy and tenodesis, and there is no consensus on which tenodesis type can offer optimal shoulder function. Importantly, several new studies, especially new high-quality RCTs were published [17,18,19], making it necessary to perform an up-to-date comparison of these two techniques. Obviously, a larger enrolment of RCTs makes high-quality subgroup analyses (tenodesis type and follow-up duration) possible.

The purpose of this meta-analysis was to provide updated evidence comparing tenotomy and tenodesis in patients with LHBT lesions. Such questions will be answered according to the meta-analysis of the RCTs: (1) Which procedure leads to better functional scores in terms of Constant scores, American Shoulder and Elbow Surgeons (ASES) scores, Simple Shoulder Test (SST) score, etc.? (2) Which approach leads to greater functional improvement after surgery? (3) Do both surgeries have similar effects on pain relief, range of motion and muscle strength? (4) Which tenodesis type is associated with optimal shoulder function?

## 2. Methods

This review was conducted under the instruction of the Cochrane Handbook for Systematic Reviews of Interventions [20] and reported based on the Preferred Reporting Items for Systematic Reviews and Meta-analyses (PRISMA) checklist [21]. Two independent reviewers (ZC, YG) conducted the literature search, study selection and literature assessment, with divergent opinions solved by debate or by further discussion with the third senior researcher (TX).

## 3. Search Strategy

A systematic search of the literature in PubMed, Cochrane Library, Embase and Web of Science was performed on 12 January 2022. The key words were arranged as follows: (long head OR biceps OR biceps tendon OR long head of the biceps tendon OR LHB OR LHBT) AND (tenodesis OR tenodeses OR tenodesing OR tenodesed) AND (tenotomy OR tenotomies OR tendon release OR release tendon OR releases tendon OR tendon releases).

## 4. Study Selection

Titles and abstracts were screened after the literature search, followed by full text assessment for eligibility. Studies were considered eligible if (1) the patients with LHBT lesions underwent biceps tenotomy or tenodesis; (2) they were RCTs; (3) they directly compared the clinical or functional outcomes of tenotomy and tenodesis; and (4) they were published in English. No restrictions on follow-up duration, year of publication or number of patients were applied. Exclusion criteria: (1) nonhuman studies, (2) case reports, case series, cohort studies or letters to the editor, and (3) incompletely published literature.

## 5. Outcomes

### Outcomes Reported by at Least Three Studies Were Summarised

The constant scores at the short- and mid-term follow-up (≤12 months and >12 months respectively, defined by the included studies [22,23,24]), and different tenodesis types (intracuff, subpectoral or suprapectoral tenodesis) were also analysed. Intracuff tenodesis was defined according to the definition by Cho NS et al. [25]. The tenotomised biceps tendon was sutured under the rotator cuff, thereby making the long head of the biceps tendon contact the undersurface of the articular portion of the rotator cuff.The constant score improvement, at the short- and mid-term follow-up (≤12 months and >12 months respectively) were compared with the preoperative baseline scores. Similarly, the constant score improvement of different tenodesis types was also analysed.Visual analogue scale (VAS) score, range: 0 to 10.American Shoulder and Elbow Surgeons (ASES) score and its improvement.Simple Shoulder Test (SST) score and its improvement.Popeye Deformity.Cramping Bicipital Pain.The elbow strength index (ESI) was defined as the ratio of elbow flexion strength (recorded with kg, N, N·m or lb) of the surgery side and contralateral side. If both the maximum and average strength were tested, the maximum strength data were extracted.The forearm supination strength index (FSSI) was defined as the ratio of forearm supination strength (recorded with kg, N, N·m or lb) of the surgery side and contralateral side, with the maximum strength pooled for analysis.Range of motion (ROM): external rotation with the arm at the side.

## 6. Data Extraction

The primary data extraction was completed by one reviewer (ZC) in accordance with the Cochrane recommendations, including evidence lever, study region, number of patients (shoulders) analysed, mean age, mean follow-up duration, tenodesis type and all clinical outcomes. The mean difference (MD) and standard deviation (SD) were recorded for continuous data, while events and total were recorded for dichotomous data. If the 95% confidence interval (CI) was reported in the article, SD was calculated under Cochrane instructions; if only data of full range and mean difference were shown, SD was not calculated due to the proven instability of the range and inaccuracy of estimation [20]. The mean differences and standard deviations of improvement of Constant scores, ASES score and SST score from baseline were inputted with the generally accepted method described by Follmann et al. and Abrams et al. [26,27]. The other reviewer (YG) revised the data.

## 7. Evaluation of the Risk of Bias

The risk of bias was assessed by two independent reviewers (ZC, YG), with a consensus reached on discrepancies. The Cochrane tool (RoB 2) [28] was used in the assessment of randomised controlled studies. Seven domains were assessed: randomisation process, allocation concealment, blindness of participants, blindness of outcome measurement, incomplete outcome report and other biases.

## 8. Statistical Analysis

The results were reported if the outcome was used in 3 or more RCTs. Forest plots were generated for outcomes reported by 3 or more RCTs.

The analyses were completed using RevMan (version 5.4.1, the Cochrane Collaboration). MD with 95% CI for continuous data and OR with 95% CI for dichotomous data of each study were calculated. Heterogeneity was tested with *I*^2^ and the chi-squared metric. Meta-analysis was performed with a fixed-effect model when *I*^2^ ≤ 50% and a random-effect model was applied when *I*^2^ > 50%. A significant *p* value was set as <0.05.

## 9. Results

### Study Characteristics

We found 1104 records in the database search. After duplicate removal, we screened 655 records, from which we reviewed 37 full-text documents. We excluded 27 studies from our review. The reasons for exclusion are listed in Figure 1. Finally, 10 studies met the inclusion criteria (Figure 1). These 10 RCTs directly compared the clinical results of tenotomy and tenodesis for LHBT lesions, including a total of 787 participants (390 tenotomy and 397 tenodesis). The characteristics of all included studies are summarised in Table 1.

## 10. Risk of Bias Assessment

The risk of bias of RCTs was assessed with RoB2 and is shown in Figure 2. Two studies [23,33] did not clearly explain how patients were allocated into two groups. Mardani et al. [23] did not blind the patients during the trial, while three other studies [22,32,33] did not mention blinding of patients. Three studies [22,23,29] did not blind assessors during outcome assessment, whereas two studies [30,33] did not report whether the outcome assessment was blinded. Two studies [18,30] reported that more than 10% of patients were lost to final follow-up, but explained and addressed the problem properly. Lee et al. [31] missed the standard deviations of all functional scores at the final follow-up. Castricini et al. [30] did not report the ROMs of forward flexion, abduction and internal rotation.

### 10.1. Constant Scores

Constant scores were reported in six RCTs. The meta-analysis of six RCTs (223 tenotomy and 220 tenodesis) showed higher Constant scores in the tenodesis group (MD, −1.24 [95% CI, −2.00 to −0.48]; *p* = 0.001) (Figure 3). Subgroup analyses of tenodesis type (intracuff, subpectoral or suprapectoral tenodesis) and follow-up duration (short-term or mid-term follow-up) were performed. Constant scores were significantly higher in the intracuff tenodesis subgroup (*p* = 0.01) and significantly higher in both the short-term (*p* = 0.008) and long-term (*p* = 0.004) follow-ups when compared with tenotomy (Table 2).

### 10.2. Constant Score Improvement

Constant score improvement was imputed in six RCTs that reported both the MD and SD of Constant scores at baseline and at the final follow-up. The meta-analysis of six RCTs (223 tenotomy and 220 tenodesis) showed greater Constant score improvement in the tenodesis group (MD, −1.54 [95% CI, −3.04 to −0.05]; *p* = 0.04) (Figure 4). Similar subgroup analyses were also performed for Constant score improvement (Table 3) and showed more significant improvement in Constant scores in the intracuff tenodesis group (*p* = 0.001).

### 10.3. VAS for Pain

The VAS score for pain was reported in five RCTs. The meta-analysis of five RCTs (232 tenotomy and 225 tenodesis) did not show a significant difference between the two groups (MD, 0.08 [95% CI, −0.21 to 0.37]; *p* = 0.59) (Figure 5).

### 10.4. ASES and SST Scores

The ASES score was reported in three RCTs. The meta-analysis of three RCTs (90 tenotomy and 88 tenodesis) did not show a significant difference between tenotomy and tenodesis (MD, −3.51 [95% CI, −12.00 to 4.98]; *p* = 0.42) (Figure 6A). The analysis of improvement in ASES score did not reveal superiority of any procedure (MD, −0.51 [95% CI, −9.58, 8.56]; *p* = 0.91) (Appendix A).

The SST score was reported in three RCTs. The meta-analysis of three RCTs (70 tenotomy and 77 tenodesis) showed significantly higher SST scores in the tenodesis group (MD, −0.73 [95% CI, −1.40 to −0.06]; *p* = 0.03) (Figure 6B). Moreover, tenodesis was associated with a larger improvement in SST score (MD, −0.76 [95% CI, −1.17, −0.34]; *p* < 0.001) (Appendix A).

### 10.5. Popeye Deformity

Popeye deformity was reported in 10 RCTs. The meta-analysis of 10 RCTs (371 tenotomy and 376 tenodesis) showed significantly less frequent Popeye deformity in the tenodesis group (OR, 3.34 [95% CI, 2.19 to 5.09]; *p* < 0.001) (Figure 7A).

### 10.6. Cramping Pain

Cramping bicipital pain was reported in three RCTs. The meta-analysis of three RCTs (133 tenotomy and 138 tenodesis) showed significantly less frequent cramping pain in patients who underwent tenodesis for LHBT (OR, 3.36 [95% CI, 1.36 to 8.27]; *p* = 0.008) (Figure 7B).

### 10.7. Elbow Flexion Strength Index

The ESI was reported or calculated in six RCTs. The meta-analysis of six RCTs (245 tenotomy and 263 tenodesis) did not show a significant difference in elbow flexion strength between the two groups (MD, 0.01 [95% CI, −0.01 to 0.03]; *p* = 0.38) (Appendix A).

### 10.8. Forearm Supination Strength Index

The FSSI was reported or calculated in four RCTs. The meta-analysis of four 4 RCTs (171 tenotomy and 186 tenodesis) did not show a significant difference in forearm supination strength between the two groups (MD, 0.04 [95% CI, −0.14 to 0.21]; *p* = 0.68) (Appendix A).

### 10.9. ROM: Shoulder External Rotation

Range of motion of shoulder external rotation was reported in three RCTs. The meta-analysis of three RCTs (102 tenotomy and 99 tenodesis) did not show a significant difference in the ROM of shoulder external rotation between the two groups (MD, −0.94 [95% CI, −4.65, 2.76]; *p* = 0.62) (Appendix A).

## 11. Discussion

In this meta-analysis, we analysed the clinical outcomes of tenotomy versus tenodesis for LHBT pathologies that were reported in RCTs only. The major findings of this meta-analysis of RCTs were that tenodesis appeared to provide more satisfactory results in terms of Constant scores, SST scores, the incidence of Popeye deformity and cramping bicipital pain after surgery. However, significant advantages favouring any procedure were not identified by assessing the VAS score for pain, ASES score, biceps strength and shoulder ROM, which was consistent with several previous reviews [10,11,34].

Different from most previous meta-analyses [10,11,12,13,34], this study revealed significantly higher Constant scores and SST scores, thus suggesting the superiority of tenodesis, which was supported by evidence from RCTs. Moreover, due to a larger number of studies, subgroup analyses of different tenodesis types and Constant scores and their improvement were considered; such subgroup analyses were not described in previous meta-analyses. The updated functional scores may have been presented in newly included RCTs [17,18,19], especially the most recent RCT conducted by van Deurzen et al. [19]. These results are new solid evidence favouring tenodesis, which appears to have a more notable effect on Constant scores and their improvement.

The optimum type of tenodesis has always been the focus of several discussions. According to the fixed mode, tenodesis is generally divided into three types: intracuff tenodesis, suprapectoral tenodesis and subpectoral tenodesis [35]. Subgroup analyses in the current study revealed significantly higher Constant scores in the intracuff tenodesis group, short-term follow-up group and mid-term follow-up group. Furthermore, all three kinds of tenodesis resulted in better results than tenotomy, which is consistent with a network meta-analysis by Anil et al. [14]. However, intracuff tenodesis presented the highest Constant scores (2.86 higher scores) and the largest improvement from baseline (5.87 higher scores). Anil et al. [14] compared suprapectoral tenodesis, intracuff tenodesis and subpectoral tenodesis with tenotomy and suggested that subpectoral tenodesis was the most superior treatment with respect to Constant scores. However, the conclusion drawn by Anil et al. should be interpreted with caution, as both RCTs and cohort studies were pooled in the network meta-analysis. The lack of studies and a small population size were major limitations for the subgroup analyses of tenodesis types in previous meta-analyses of RCTs.

To date, there has been no conclusion on the comparison of short-term and long-term effects between tenotomy and tenodesis. A recent meta-analysis of only RCTs compared the Constant scores of tenodesis and tenotomy in the short-, mid- and long-term at 4 weeks, 6 months and >1 year after surgery, respectively, and found no significant differences between groups [12].To date, this was the only meta-analysis reporting the subgroup follow-up duration. However, due to the limitation of the constituent studies, no analysis of more than 2 years of follow-up from RCTs could be performed, and the meta-analysis might be underpowered to detect significant differences. In our meta-analysis of RCTs, the short- and mid-term follow-up groups showed significantly higher Constant scores at 1 and 2 years after surgery, which suggested the superiority of tenodesis. Moreover, the cohort study conducted by Godenèche et al. [36] was the only study reporting results of more than 5 years of follow-up. It was concluded that tenodesis rendered higher Constant scores (4.50 higher scores, *p* = 0.025) than tenotomy after 10 years of follow-up. This limited evidence of the long-term result, together with our subgroup meta-analysis, suggested that tenodesis could be preferred, when optimal function of the shoulder is needed, in the short-, mid- or even long-term.

A meta-analysis of other clinical scores was also performed. Contrary to our initial hypothesis, the SST score and its improvement were significantly higher in favour of tenodesis, while the ASES score remained similar between tenotomy and tenodesis. Even though all the differences in the clinical scores were below the MCID (Constant scores: 10.4 [37], SST scores: 2.33 points [38]), they imply optimistic functions, which could be essential for certain patients including high-level athletes. Complications of cosmetic deformity and cramping bicipital pain have not always been a major concern, although they occur more often after tenotomy. Both procedures had high rates of patient-reported satisfaction [19,22,32,33,39,40], but tenodesis had a higher satisfaction rate in one study [23].

## 12. Limitations

Several limitations below were ascribed to the inherent limitations of the included studies. First, heterogeneity of surgical procedures, follow-up duration and patient age in each RCT made the application of the conclusion difficult. However, we were able to perform subgroup analyses with the aim of rendering a clearer interpretation of the results. Second, there was no study with high-level evidence reporting results of more than 5 years of follow-up, thus leaving a question mark on long-term results. Third, one RCT pooled in the meta-analysis excluded patients with a history of conservative treatment before surgery [23], which is not a proper exclusion. However, the conclusions remain unaffected whether including this study or not, leading to our decision of constitution. Fourth, only one randomised trial [17] included patients with LHBT pathologies without rotator cuff tears, and this high rate of concomitant disorders, although frequently seen in the clinic, adds complexity to the explanations of the results. Additionally, the MCIDs determined for each clinical score are based on rotator cuff tears rather than LHBT lesions [37,38]. Further studies on determining MCIDs of these clinical measurements on LHBT lesions are required, and will contribute to more precise evaluations of each pathology. Furthermore, studies that failed to be obtained in full text were not included in this study, so some valuable studies may have been omitted. Nevertheless, we are confident that none of these methodological limitations would change the overall conclusions of this review.

## 13. Conclusions

According to analyses of the RCTs, tenodesis seems to offer better functional scores and greater functional improvement after surgery when compared with tenotomy in terms of Constant scores, improvement of Constant scores, SST scores, improvement of SST scores, and the incidences of Popeye deformity and cramping bicipital pain after surgery. Significant advantages favouring any procedure were not identified in terms of VAS score for pain, ASES score, biceps strength and shoulder ROM. Intracuff tenodesis might provide the best shoulder function as measured with Constant scores. More high-quality RCTs comparing tenodesis and tenotomy with a long-term follow-up are still needed.

## Figures and Tables

**Figure 1 jcm-12-01754-f001:**
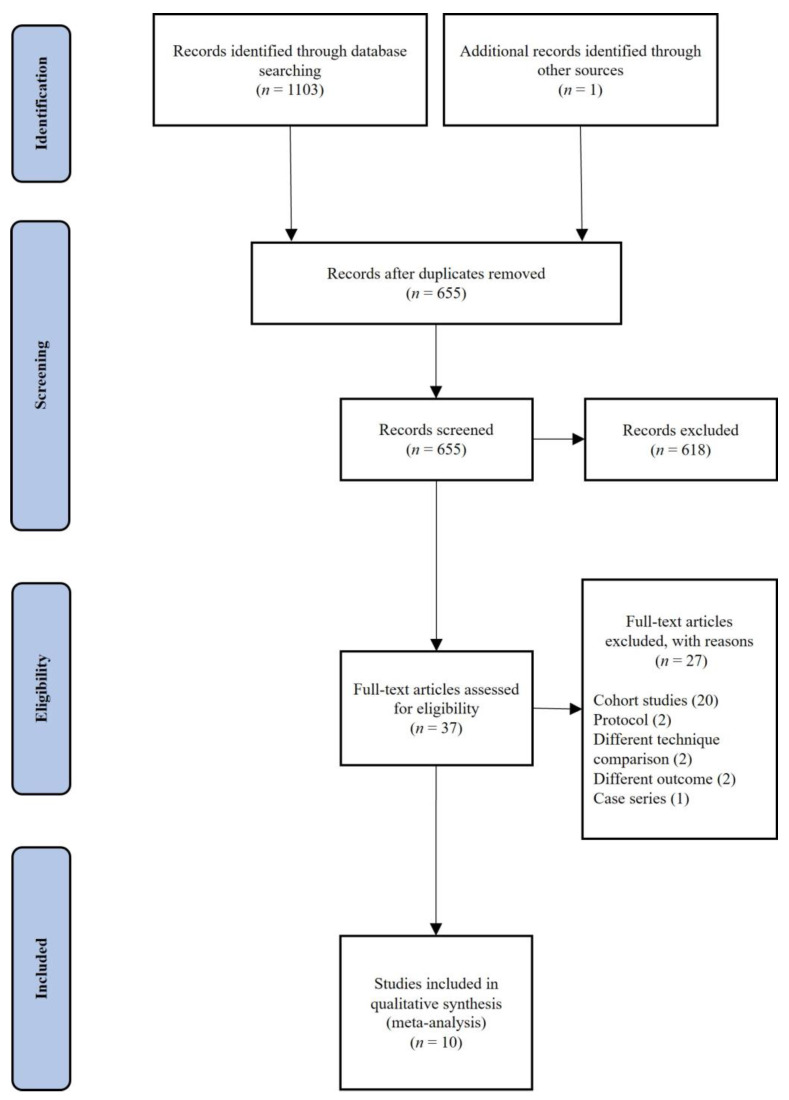
PRISMA (Preferred Reporting Items for Systematic Reviews and Meta-Analyses) flowchart of literature retrieval and selection.

**Figure 2 jcm-12-01754-f002:**
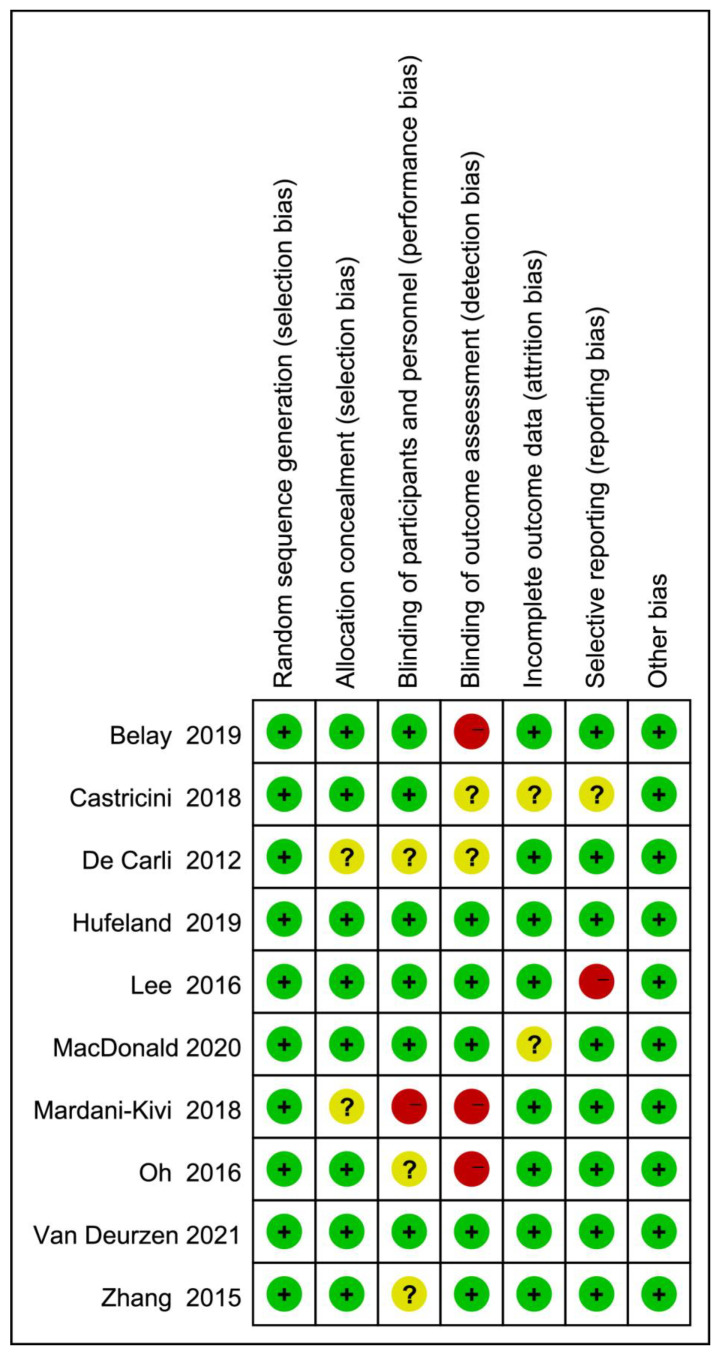
Risk of bias assessment of randomised controlled trials. Green means low risk. Red means high risk. Yellow means unclear risk [17,18,19,22,23,29,30,31,32,33].

**Figure 3 jcm-12-01754-f003:**
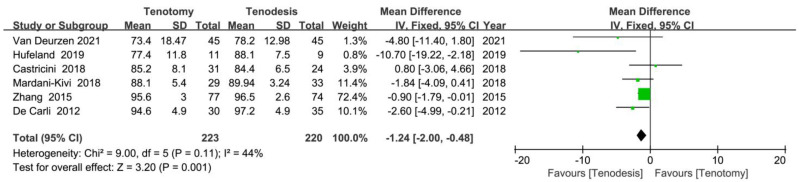
Meta-analysis of randomised controlled trials showing that Constant scores were better in the patients who underwent tenodesis. Green symbol: mean difference value; Black symbol: result of meta-analysis [17,19,23,30,32,33].

**Figure 4 jcm-12-01754-f004:**
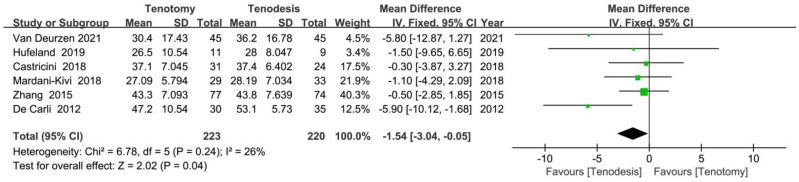
Meta-analysis of randomised controlled trials showing the Constant score improvement from baseline was greater in the patients who underwent tenodesis. Green symbol: mean difference value; Black symbol: result of meta-analysis [17,19,23,30,32,33].

**Figure 5 jcm-12-01754-f005:**
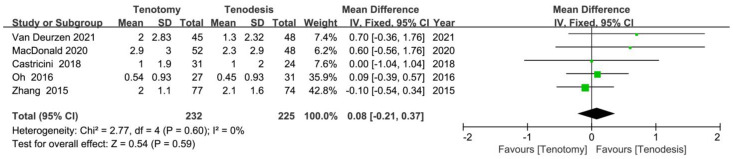
The visual analogue scale (VAS) score for pain showed no significant difference in the meta-analysis of randomised controlled trials. Green symbol: mean difference value; Black symbol: result of meta-analysis [18,19,22,30,32].

**Figure 6 jcm-12-01754-f006:**
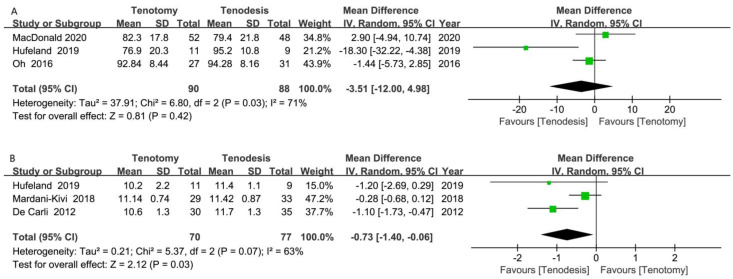
Meta-analysis of the (**A**) American Shoulder and Elbow Surgeons (ASES) score; (**B**) Simple Shoulder Test (SST) score. Green symbol: mean difference value; Black symbol: result of meta-analysis [17,18,22,23,33].

**Figure 7 jcm-12-01754-f007:**
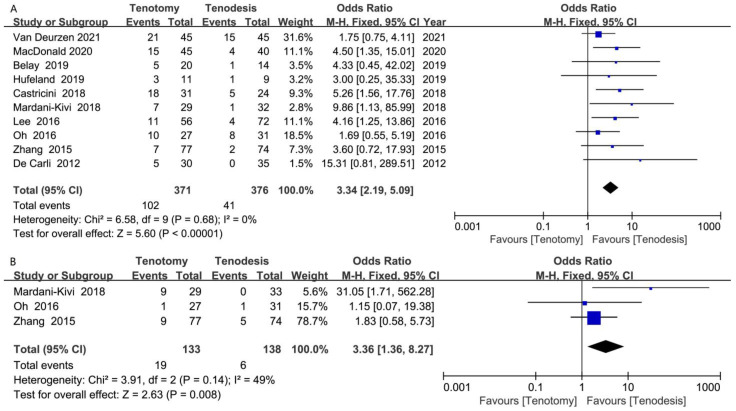
Meta-analysis of (**A**) Popeye deformity; (**B**) cramping bicipital pain. Blue symbol: odds ratio; Black symbol: result of meta-analysis [17,18,19,22,23,29,30,31,32,33].

**Table 1 jcm-12-01754-t001:** Characteristics of the included studies.

Author	Year	Study Type	LoE	Sample Size			Mean Age	Mean FU (Mon.)	Tenodesis Type	Outcomes
				Total	Tenotomy	Tenodesis				
Van Deurzen [19]	2021	RCT	I	100	52	48	61	12.3	intracuff	Constant score, DASH, popeye deformity, ESI, VAS, satisfaction
MacDonald [18]	2020	RCT	I	114	57	57	57.7	24	Subpectoral/ Suprapectoral	VAS, ASES, elbow and shoulder strength, popeye deformity
Hufeland [17]	2019	RCT	I	20	11	9	52	12	Suprapectoral	Constant score, SST, ASES, ESI, FSSI, popeye
Belay [29]	2019	RCT	II	34	20	14	56	24	Suprapectoral	VAS, ASES, cramping, popeye, groove tenderness
Mardani-Kivi [23]	2018	RCT	II	62	29	33	55	24	Subpectoral	Constant score, popeye, satisfaction
Castricini [30]	2018	RCT	I	55	31	24	58.7	24	Suprapectoral	Constant score, VAS, popeye, cramping, ROM, elbow flexion strength
Oh [22]	2016	RCT	II	58	27	31	58.89	14.64	Suprapectoral	VAS, ASES, ROM, ESI, FSSI, cramping, popeye, satisfaction
Lee [31]	2016	RCT	I	128	56	72	62.9	22.1	Suprapectoral	ASES, SST, VAS, ROM, Constant score, popeye, ESI, FSSI
Zhang [32]	2015	RCT	I	151	77	74	61	25	Suprapectoral	Constant score, VAS, ESI, FSSI, popeye, satisfaction
De Carli [33]	2012	RCT	II	65	30	35	57.8	24	intracuff	Constant score, SST, ESI, popeye
Total	-	-	-	787	390	397	59.3	21.4	-	-

FU, Follow-up; RCT, Randomised controlled trial; PCS, Prospective cohort study; RCS, Retrospective cohort study; DASH, Disabilities of the Arm, Shoulder and Hand score; ESI, Elbow Strength Index; VAS, Visual Analogue Scale; ASES score, American Shoulder and Elbow Surgeons; SST score, Simple Shoulder Test; LHB scores, long head of the biceps; ROM, range of motion; FSSI, Forearm supination strength index.

**Table 2 jcm-12-01754-t002:** Summary of subgroup analyses of Constant scores.

Subgroups	Number of RCTs	*p* Value	MD	95% CI	In Favor of
Tenodesis type					
Intracuff	2	0.01	−2.86	[−5.10, −0.61]	Tenodesis
Subpectoral	1	0.11	−1.84	[−4.09, −0.41]	-
Suprapectoral	3	0.37	−1.68	[−5.35, 1.99]	-
Follow-up duration					
Short-term	2	0.008	−7.01	[−12.23, −1.79]	Tenodesis
Mid-term	4	0.004	−1.12	[−1.89, −0.35]	Tenodesis
Total	6	0.001	−1.24	[−2.00, −0.48]	Tenodesis

RCTs: randomised controlled trials; MD, mean difference; CI, confidence interval; “-”, no significant difference identified.

**Table 3 jcm-12-01754-t003:** Summary of subgroup analyses of Constant score improvement.

Subgroups	Number of RCTs	*p* Value	MD	95% CI	In Favor of
Tenodesis type					
Intracuff	2	0.001	−5.87	[−9.50, −2.25]	Tenodesis
Subpectoral	1	0.50	−1.10	[−4.29, 2.09]	-
Suprapectoral	3	0.61	−0.50	[−2.41, 1.41]	-
Follow-up duration					
Short-term	2	0.15	−3.95	[−9.29, 1.39]	-
Mid-term	4	0.09	−1.34	[−2.89, 0.22]	-
Total	6	0.04	−1.54	[−3.04, −0.05]	Tenodesis

RCTs: randomised controlled trials; MD, mean difference; CI, confidence interval; “-”, no significant difference identified.

## Data Availability

Not applicable.

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
