# Peer review of "Biceps Tenodesis Better Improves the Shoulder Function Compared with Tenotomy for Long Head of the Biceps Tendon Lesions: A Meta-Analysis of Randomised Controlled Trials"

_jcm, 2023, doi:10.3390/jcm12051754_

Round 1

Reviewer 1 Report

Despite the topic is interesting, the review presents major flaws.

The paper is very disorganized and difficult to follow.

I would include only studies in English, from major databases (eg WoS, Pubmed,..)

How was quality of studies assessed? This should be reported in results.

Which studies were included in metaanalysis?

Table 1 is not readable

Results are very confusing. Should be reorganized in sub-paragraphs. 

Discussion: confusing and disorganized.

Reviewer 2 Report

This review manuscript entitled "Biceps tenodesis better improves the shoulder function compared with tenotomy for 1 long head of the biceps tendon lesions: A systematic review and meta-analysis" focused an interesting topic regarding the biceps tenotomy and tenodesis. This topic has been debated for many years, and most reports indicate no difference in clinical outcomes. Since tenodesis has a cosmetic problem and may affect muscle weakness in younger generation, there are really few obvious indications for the LHB pathology. Regarding this point of the view, this review is very meaningful. Overall, it is well organized, but there are some questions and points to be corrected, so please consider them.

The authors mention the term “intracuff tenodesis”. According to the manuscript of Cho NS et al., this is the arthroscopic procedure of intraarticular repair which tenotomized biceps is sutured to the rotator cuff tendon. A description of this procedure should be included in the manuscript.

At the last part of the introduction, with the questions of which procedure, approach, or type of tenodesis is better, the authors describe the purpose of this meta-analysis for LHB treatment, but they do not seem to answer them clearly in their conclusions. The reader will want to know more detail about the actual procedure and which method is best.

Please describe the title of the supplement data. In particular, it is difficult to understand the difference between supplement data 3 and 4.

Round 2

Reviewer 1 Report

Thank you for the attempt of ameliorating the paper.

I was really ameliorated.

Please have it further checked for language issues. I would appreciate if the Authors could provide a certification for language check.
